# SUMOylation Is Required for ERK5 Nuclear Translocation and ERK5-Mediated Cancer Cell Proliferation

**DOI:** 10.3390/ijms21062203

**Published:** 2020-03-23

**Authors:** Tatiana Erazo, Sergio Espinosa-Gil, Nora Diéguez-Martínez, Néstor Gómez, Jose M Lizcano

**Affiliations:** Institut de Neurociències and Departament de Bioquímica i Biología Molecular, Facultat de Medicina, Universitat Autònoma de Barcelona, 08193 Bellaterra, Barcelona, Spain; erazoani@mskcc.org (T.E.); sergio.espinosa@uab.cat (S.E.-G.); nora.dieguez@uab.cat (N.D.-M.); nestor.gomez@uab.cat (N.G.)

**Keywords:** MAP kinase, ERK5, Bmk1, SUMO, nuclear translocation, transcription, cell proliferation, cancer, Hsp90, Cdc37

## Abstract

The MAP kinase ERK5 contains an N-terminal kinase domain and a unique C-terminal tail including a nuclear localization signal and a transcriptional activation domain. ERK5 is activated in response to growth factors and stresses and regulates transcription at the nucleus by either phosphorylation or interaction with transcription factors. MEK5-ERK5 pathway plays an important role regulating cancer cell proliferation and survival. Therefore, it is important to define the precise molecular mechanisms implicated in ERK5 nucleo-cytoplasmic shuttling. We previously described that the molecular chaperone Hsp90 stabilizes and anchors ERK5 at the cytosol and that ERK5 nuclear shuttling requires Hsp90 dissociation. Here, we show that MEK5 or overexpression of Cdc37—mechanisms that increase nuclear ERK5—induced ERK5 Small Ubiquitin-related Modifier (SUMO)-2 modification at residues Lys6/Lys22 in cancer cells. Furthermore, mutation of these SUMO sites abolished the ability of ERK5 to translocate to the nucleus and to promote prostatic cancer PC-3 cell proliferation. We also show that overexpression of the SUMO protease SENP2 completely abolished endogenous ERK5 nuclear localization in response to epidermal growth factor (EGF) stimulation. These results allow us to propose a more precise mechanism: in response to MEK5 activation, ERK5 SUMOylation favors the dissociation of Hsp90 from the complex, allowing ERK5 nuclear shuttling and activation of the transcription.

## 1. Introduction

The extracellular-signal-regulated kinase 5 (ERK5, also called Big MAP Kinase-1 (BMK1)) is the most structurally divergent member of the mitogen-activated protein kinase (MAPK) family. ERK5 protein contains an N-terminal kinase domain that shares 50% identity with ERK1/2 [1] and a unique C-terminal tail, with no homology with any other protein, containing a nuclear localization signal (NLS) and a transactivation domain (TAD) [2]. ERK5 activates transcription by either phosphorylation of transcription factors such as MEF2 [3], Sap1 [4] or c-Myc [5,6] or through the TAD domain in a kinase-independent mechanism.

ERK5 is ubiquitously expressed in numerous tissues and is activated in response to growth factors and to different forms of stress. ERK5 is activated by direct phosphorylation of the Thr218 and Tyr220 (TEY motif) within the activation loop of the kinase domain by the MAPK kinase 5 (MEK5) [7]. MEK5 is the only upstream kinase described for ERK5, and, therefore, MEK5-ERK5 constitutes a unique signaling axis to control cell differentiation, proliferation and survival [8]. Importantly, ERK5 plays a major role in regulating cell cycle progression and proliferation in response to a wide range of growth factors such as epidermal growth factor (EGF) [3], vascular endothelial growth factor (VEGF) [9], fibroblast growth factor-2 (FGF-2) [10], nerve growth factor (NGF) [11], colony-stimulating factor 1 (CSF-1) [12], interleukin 6 (IL-6) [13] and platelet-derived growth factor (PDGF) [14]. Mechanistically, ERK5 controls the G1/S phase transition via the transcriptional regulation of cyclin D1 [15]. Active ERK5 phosphorylates and activates MEF2A, C and D transcription factors [3,5,16], promoting c-Jun- and c-Fos-mediated expressions of cyclin D1 required for cell proliferation [4,17]. Moreover, ERK5 also allows cancer cells to evade cell cycle suppressors by impairing the expression of the CDK inhibitors p15, p21 and p27 [18,19].

In addition to the well-established role of ERK5 in sustaining proliferative signals and evading growth suppression, numerous studies have underlined an implication of ERK5 in almost all hallmarks of cancer (recently reviewed in [20]). For instance, and among others, ERK5 pathway is associated with increased metastatic activity in prostate [21] and hepatocarcinoma [19] cancers, to elevated growth of breast cancer cells, overexpressing the ErbB2/Her2 receptor [22] or to chemoresistance of breast cancer [23] or acute myeloid leukemia cells [24]. Consequently, genetic and pharmacologic manipulation of the MEK5-ERK5 pathway has a deep impact in cell and tumor viability, and, therefore, this pathway has been proposed as a new target to tackle different solid and blood cancers.

The subcellular distribution of ERK5 is critical for regulating its role in cancer proliferation and survival. Indeed, nuclear ERK5 correlates with poor prognosis in aggressive prostatic carcinoma [21,25] and in hepatocellular carcinoma [19] and also confers resistance to TRAIL-induced apoptosis in breast cancer models [26]. Thus, it is important to define the precise molecular mechanisms implicated in ERK5 nucleo-cytoplasmic shuttling. Initially, it was proposed that ERK5 N-terminal and C-terminal halves interact, forming either a putative NES or a domain that binds a cytoplasmic anchor protein [27]. After activation, the autophosphorylation of the C-terminal tail (at residues Ser421, Ser433, Ser496, Ser731 and Thr733 [28]) would disrupt the intramolecular interaction, resulting in the exposition of the NLS and nuclear translocation of ERK5 [27]. Later, we described that the chaperone Hsp90 acts as the cytosolic anchor protein. Indeed, ERK5 interacts with chaperones Hsp90 and Cdc37 in basal conditions, which stabilize and retain ERK5 at the cytosol [29]. In a canonical mechanism, and in response to mitogenic factors, MEK5-mediated ERK5 activation results in phosphorylation of the C-term tail, Hsp90 dissociation and nuclear shuttling. In a noncanonical mechanism, the overexpression of Cdc37 (as it happens in several cancers [30]) also induces Hsp90 dissociation and nuclear translocation of a catalytically inactive—but transcriptionally active—form of ERK5 [29].

Small Ubiquitin-related Modifier (SUMO) is a type of post-translational modification that plays an important role in regulating protein activity, stability, interactions with other proteins and subcellular localization, among others [31]. SUMO is covalently linked to lysine residues in the target proteins through a process that is mechanistically analogous to ubiquitination. The SUMO pathway begins with a SUMO-activating enzyme (SAE1/SAE2), which induces the activation of the SUMO C-terminal domain and transfers activated SUMO to the SUMO-conjugating enzyme (SUMO E2 ligase) Ubc9 [32]. Finally, one of several SUMO-protein E3 ligases PIAS (protein inhibitor of activated STAT) promotes the covalent binding between the lysine of the protein target and the glycine-glycine dipeptide at the SUMO C-terminus. SUMO modification is a reversible and transient protein modification, and it is de-conjugated from targets by specific SUMO isopeptidases called sentrin-specific proteases (SENP) [33].

In this study, we aimed to investigate the role of SUMO modification in the nuclear import of ERK5. We show that ERK5 becomes SUMOylated in response to EGF-induced MEK5 phosphorylation and Cdc37 overexpression and that SUMOylation is a necessary event but not sufficient, per se, for ERK5 nuclear shuttling.

## 2. Results

### 2.1. MEK5 Activity and Cdc37 Overexpression Induce ERK5 SUMOylation at Lys6 and Lys22

It has been previously reported that advanced glycation end products (AGE) and H2O2 induce ERK5 SUMOylation at residues Lys6 and Lys22 in cultured endothelial cells and cardiomyocytes and that this posttranslational modification modulates the transcriptional activity of ERK5 [34,35]. However, the authors did not explore the relevance of SUMO modification in ERK5 cellular localization. Given the role of SUMOylation in the nuclear translocation of targeted proteins, we decided to study the effect of this posttranslational modification on the subcellular localization of ERK5 in cancer cells. To do so, we have used two different paradigms for ERK5 nuclear shuttling: active MEK5 (canonical mechanism) and Cdc37 (noncanonical mechanism) overexpression [36].

To investigate whether SUMO modification is required for ERK5 nuclear translocation, we performed in vivo SUMOylation assays. We transiently overexpressed in HEK293T cells SUMO2, the E2-conjugating enzyme Ubc9 and FLAG-tagged ERK5, in combination with a constitutive active form of MEK5 (MEK5DD, in which residues Ser311 and Ser315 have been replace by Asp or Cdc37, to induce ERK5 nuclear translocation. Levels of SUMOylated ERK5 were determined by immunoprecipitating ERK5 and immunoblotting for SUMO2. Figure 1A shows that overexpression of MEK5DD or Cdc37 resulted in increased levels of slow migrating species of ERK5. SUMO2 contains an acceptor Lys (Lys11) that allows the formation of polySUMO chains [32]. Therefore, the high-molecular-weight smear of bands corresponded to SUMOylated ERK5 species. We also performed experiments with HEK293T cells overexpressing His-tagged SUMO2, Ubc9 and GST-tagged ERK5 wild type or an ERK5 SUMO-deficient mutant (K6/22R, in which the two SUMO sites Lys6 and Lys22 have been replaced by Arg. Levels of SUMOylated ERK5 were determined by pulling down His-SUMO2 from lysates with Ni^2+^-NTA-agarose beads and immunoblotting for either GST or ERK5 proteins. Overexpression of MEK5DD or Cdc37 resulted in slow migrating species of wild type ERK5 but not of the ERK5-SUMO-deficient mutant, confirming that Lys 6 and 22 are the residues that covalently bind SUMO (Figure 1B,C). Parallel experiments overexpressing a mutant form of ERK5 that lacks a C-terminal tail (ERK5(1-490)) showed SUMO conjugation in response to MEK5DD or Cdc37 overexpression, indicating that the C-term tail is not required for the interaction with the putative SUMO E3 ligase (Figure 1D). These results demonstrate for the first time that MEK5 activity or Cdc37 overexpression induce ERK5 SUMOylation in Lys6 and Lys22 residues.

### 2.2. SUMOylation Is Required for ERK5 Nuclear Translocation in Response to MEK5-Mediated Activation or to Cdc37 Overexpression

To study the role of SUMOylation in ERK5 nuclear translocation, we used human prostatic adenocarcinoma PC-3 cells, a good cellular model to study the cytoplasmic-nuclear trafficking of ERK5 [21,29]. We overexpressed either ERK5 or the ERK5-SUMO deficient mutant ERK5-K622/R, and ERK5 subcellular localization was monitored by either fluorescence microscopy (Figure 2A) or subcellular fractionation (Figure 2B). As expected, overexpressed ERK5-WT showed cytosolic localization in basal conditions (as it does endogenous ERK5) and nuclear staining in response to overexpression of MEK5DD or Cdc37. In contrast, the SUMO-deficient mutant did not translocate to the nucleus in response to overexpression of any of these two proteins. Furthermore, 18-h incubation of cells with the inhibitor of nuclear protein export 20-nM Leptomycin B resulted in nuclear accumulation of ERK5-WT but still did not affect cytosolic localization of ERK5-SUMO-deficient mutant (Figure 2C). These results show for first time that SUMOylation is required for ERK5 nuclear translocation in response to MEK5-mediated activation or to Cdc37 overexpression.

We performed analogous experiments to study the subcellular localization of endogenous ERK5. Several authors have previously shown that epidermal growth factor (EGF) induces ERK5 activation and nuclear translocation in different cancer cells [3,19,27]. As expected, PC-3 cells showed cytosolic ERK5 in basal conditions and nuclear localization in response to short-time EGF stimulation (15 min) (Figure 3A). Protein SUMOylation could be reverted by the sentrin-specific proteases (SENPs), which break the linkage between the N-terminal Gly of SUMO and the Lys of the targeted protein [33,37]. Thus, we next investigated the role of ERK5 SUMOylation in EGF-induced nuclear translocation by reverting its SUMOylation using the SUMO protease SENP2, the isoform involved in ERK5 deSUMOylation [38]. Overexpression of SENP2 in PC-3 cells prevented endogenous ERK5 nuclear translocation in response to EGF stimulation or Cdc37 overexpression (Figure 3B,C, respectively), indicating that SUMOylation is also necessary for the nuclear translocation of endogenous ERK5.

### 2.3. SUMO Modification Does not Affect ERK5 Kinase Activity or Its Activation by MEK5

Next, we explored whether SUMOylation affects ERK5 kinase activity. To do so, we performed in vitro kinase assays using HEK293T cell lysates overexpressing GST-tagged ERK5, alone or in combination with active MEK5, and ^32^P-ATP and PIMtide peptide as substrates [29]. MEK5DD overexpression resulted in a robust increase of kinase activity of both ERK5-WT and SUMO-deficient mutant (Figure 4A). Moreover, ERK5 WT and kinase-inactive mutant (D200A) resulted in SUMOylated in response to overexpression of active MEK5 or Cdc37 (Figure 4B). These results indicate that not only ERK5 kinase activation does not require SUMOylation but also that SUMOylation is independent of ERK5 kinase activity. Our observations are in agreement with those reported by Woo et al. who showed that SUMOylation in response to H2O2 or AGE does not affect ERK5 kinase activity [34]. Interestingly, we observed that the ERK5 kinase-inactive mutant did not translocate to the nucleus in response to MEK5 overexpression (Figure 4C), indicating that SUMOylation is not sufficient, per se, to induce ERK5 nuclear translocation.

### 2.4. SUMOylation Is Required for ERK5 Co-Transcriptional Activity and Cell Proliferation

Nuclear ERK5 acts as an activator of different transcription factors, such as MEF2 or AP-1 transcriptional complex [17]. Given our results showing that ERK5 nuclear translocation requires SUMOylation, we undertook the corresponding transcriptional assays to monitor AP-1 transcriptional activity in PC-3 cells overexpressing ERK5 WT or SUMO-deficient mutants in a luciferase-based reporter assay. As previously reported [29], active MEK5 or Cdc37 overexpression resulted in enhanced ERK5-mediated AP-1 transcriptional activity. In turn, the SUMO-deficient mutant had negligible effect on AP-1 transcriptional activity in response to MEK5 or Cdc37 overexpression (Figure 5A), indicating that ERK5 SUMOylation is necessary for ERK5-mediated AP-1 transcriptional activity.

Of note, ERK5 plays an important role on the proliferation of cancer cells through activating transcriptional activity of pro-proliferative factors such as AP-1 complex, among others. Therefore, we next asked whether ERK5 requires SUMO modification to activate cell proliferation. We used PC-3 prostatic cancer cells, since overexpression of MEK5 or ERK5 results in enhanced proliferation of these cells [21,25,39]. In agreement with our results showing that SUMOylation is required for ERK5-mediated AP-1 activity, ERK5 overexpression resulted in an increase on the PC-3 cell proliferation index, whereas no effect was observed after overexpressing the SUMO-deficient ERK5 mutant. Furthermore, overexpression of ERK5, but not of the SUMO-deficient ERK5 mutant, also enhanced cell proliferation induced by active MEK5 (Figure 5B,C). Overall, our results suggest that SUMO modification plays a role in modulating ERK5-mediated cancer cell proliferation, which agrees with reports showing that SUMO modification of key proteins results in enhanced cancer cell proliferation [40].

### 2.5. SUMOylation Is Necessary for Hsp90 Dissociation from the ERK5-Cdc37 Complex

We previously showed that Hsp90 dissociation from the ERK5-Cdc37 complex is a necessary event for ERK5 nuclear translocation, since Hsp90 acts as a cytosolic anchor for ERK5 [29]. Given our results showing that SUMO modification is also necessary for ERK5 shuttling to the nucleus, we next investigated whether this posttranslational modification is also involved in Hsp90 dissociation. To this end, we transiently expressed in HEK293T HA-tagged Hsp90 and GST-tagged ERK5 WT or the ERK5 SUMO-deficient mutant cells, and ERK5 was activated by overexpression of active MEK5 (MEK5DD). Then, we analyzed the presence of these proteins in the affinity-purified ERK5 or Hsp90 pellets. As reported before, activation of ERK5 by MEK5 resulted in dissociation of Hsp90 from ERK5 WT (Figure 6). However, overexpression of active MEK5 did not induce the release of Hsp90 from the ERK5 SUMO-deficient complex. These results suggest that SUMO modification is a necessary event for Hsp90 dissociation in response to MEK5 activation.

## 3. Discussion

ERK5 controls proliferation of cancer cells by acting as a transcriptional co-activator at the nucleus [3,13,25,41]. Therefore, it is important to establish the precise molecular mechanisms involved in ERK5 nuclear shuttling, not fully described yet. A previous work showed that ERK5 is SUMOylated at Lys6 and Lys22 in endothelial cells in response to different stresses [34,35], but the authors did not investigate whether this posttranslational modification affects ERK5 transport to the nucleus. Here, we show that MEK5 or Cdc37 overexpression—two mechanisms that induce nuclear ERK5—results in ERK5 SUMOylation at residues Lys6 and Lys22 in cancer cells (Figure 1). We found that SUMO modification does not affect ERK5 kinase activity but that it is absolutely required for ERK5 nuclear translocation, since SUMO-deficient ERK5 mutant showed a constitutive cytoplasmic localization even in the presence of the nuclear export inhibitor leptomycin B (Figure 2 and Figure 3). Supporting this notion, overexpression of the SUMO protease SENP2 (the enzyme catalyzing ERK5 de-SUMOylation, [37]) completely abolished endogenous ERK5 nuclear localization in response to EGF stimulation (Figure 3).

SUMO modification plays an important role in regulating the subcellular localization of many target proteins. In mammalian cells, protein SUMOylation has been linked to nuclear import [31], and most of the SUMO machinery enzymes are found within the nucleus. That is the case for the IGF-1 receptor, which becomes SUMOylated after IGF-1 stimulation and translocates to the nucleus in a SUMO-dependent process [42], the soluble intracellular domain of the ErbB4 receptor [43] or the Polo-like kinase 1 [44], among others. These kinases do not possess an NLS motif, but SUMO modification seems to allow them to interact with nuclear transport proteins, probably facilitating the binding to cargo proteins containing NLS sequences. In the case of ERK5, which possesses a hidden NLS motif, we cannot discard that SUMO modification could facilitate its binding to a cargo protein. However, it is also probably that branched SUMOylation could result in a conformational change of ERK5 into an open conformation that would expose the NLS motif, allowing its transport to the nucleus.

The results reported here allow us to propose a more precise mechanism for the nuclear transport of ERK5 in response to MEK5-mediated canonical activation (summarized in Figure 7). This model integrates our previous findings about the role of the cytosolic anchor Hsp90 protein, together with posttranslational modifications such as phosphorylation and SUMOylation. Thus, in basal conditions, the C-terminal tail of ERK5 interacts intramolecularly with the N-terminal half, generating a close conformation that enables interaction with the cytosolic anchor protein Hsp90 and Cdc37 co-chaperone. Mitogens such as EGF induce the activation of the upstream kinase MEK5, which in turn activates ERK5 by phosphorylating the TEY motif. This phosphorylation allows the recruitment of a specific PIAS SUMO E3 ligase (yet to be described), resulting in SUMOylation of ERK5 (branched SUMO chains) at residues Lys6 and Lys22. Of note, our findings show an interplay between phosphorylation and SUMOylation, as described for other proteins (reviewed in [45]). In this model, covalent SUMO modification of ERK5 would facilitate the dissociation of Hsp90 from the kinase domain (Hsp90 is not dissociated from the ERK5 SUMO-deficient mutant in response to MEK5 activation, Figure 6). Then, active ERK5 autophosphorylates its C-term tail, inducing a conformational change that results in exposure of the NLS and in the transport of ERK5 to the nucleus. In our model, SUMOylation is a necessary event but not sufficient, per se, for MEK5-mediated ERK5 nuclear translocation, since the ERK5 kinase inactive mutant (D200A) becomes SUMOylated in response to MEK5 activation (Figure 4B), but it does not translocate to the nucleus (Figure 4C).

We previously described that overexpression of Cdc37 also results in Hsp90 dissociation and ERK5 nuclear shuttling. This mechanism does not require ERK5 kinase activity, since Cdc37 overexpression induces nuclear translocation of the kinase dead mutant D200A or the wild type ERK5, even in the presence of a XMD8-92 inhibitor [29]. Here, we show that Cdc37 overexpression results in SUMOylation of the ERK5-D200A mutant (Figure 4B). Thus, it is likely that binding of the overexpressed Cdc37 would induce a conformational change in ERK5, independently of its kinase activity and similar to that induced by the autophosphorylation of the C-terminal tail. This change, then, would facilitate the recruitment of the putative SUMO ligase, resulting in SUMOylation and the nuclear shuttling of ERK5. Further experiments will be required in order to identify the domain of ERK5 involved in the binding of the SUMO ligase and to shed light into the precise mechanism involved in ERK5 nuclear translocation.

Our results indicate that SUMOylation is necessary for the transcriptional activity of ERK5, since the SUMO-deficient mutant did not show AP-1 transcriptional activity (Figure 5C). Nevertheless, this can lead to misinterpretation of the real role of SUMO modification on ERK5 transcriptional activity. Given the fact that SUMOylation is required for ERK5 nuclear import and that ERK5 needs to be at the nucleus to act as transcriptional co-activator, it can be expected that the SUMO-deficient mutant will not show transcriptional activity. However, it cannot be ruled out that ERK5 could be de-SUMOylated by the SENP2 protease at the nucleus. SENP2 localizes at the internal side of the nuclear pore via interaction with the nuclear pore protein NUP153 [46], and it is involved in the de-SUMOylation of proteins after nuclear entry [42]. Supporting this, we found that SENP2 specifically repressed endogenous ERK5 nuclear translocation induced by EGF stimulation (Figure 3C). Thus, we cannot discard that ERK5 is subjected to a dynamic SUMOylation/de-SUMOylation cycle at the nuclear pore and, therefore, that nuclear de-SUMOylation is required for ERK5 to act as a transcriptional co-activator. In this regard, it has been previously shown in nontumoral endothelial cells that advanced glycation end products (AGE) and H2O2 induce ERK5 SUMOylation at Lys6 and Lys22, resulting in the impaired ERK5-mediated transcriptional activity of MEF2 transcription factor in response to MEK5 overexpression [34,35]. Thus, these observations suggest that non-SUMOylated ERK5 is the species that interacts with transcription factors. Further experiments will be required to address this important question.

Finally, and in line with data presented here indicating that SUMOylation is required for ERK5 nuclear translocation, we show that SUMO modification is necessary for ERK5-induced AP-1-mediated proliferation of prostatic adenocarcinoma PC-3 cells (Figure 5B,C). This result is in line with those reporting the relevance of nuclear ERK5 in cancer cell proliferation (reviewed in [36,47]). Expression of key components of the SUMO machinery (such as SUMO1 E1 activating enzyme, E2 conjugation enzyme Ubc9 and some PIAS SUMO E3 ligases) is enhanced in many human cancers, and it correlates with poor prognosis [48]. Of relevance for our work, it has been reported elevated expression levels of Ubc9 and PIAS1 enzymes in human prostate adenocarcinoma [49,50]. Given the fact that nuclear ERK5 is associated with proliferation, invasion and bad prognosis in prostate cancer [21,25], it is likely that this scenario of enhanced SUMOylation dynamics could result in favoring ERK5 SUMOylation and, therefore, ERK5 nuclear localization. In this context, it will be interesting to explore if targeting ERK5 SUMOylation could result in an effective approach to tackle prostate cancer, as well other cancers showing nuclear ERK5 and enhanced SUMO machinery players.

## 4. Materials and Methods

### 4.1. Materials

N-ethyl-maleimide, Ni2+-NTA-agarose, anti-HA-agarose, epidermal growth factor, leptomycin B and phosphocellulose paper p81 were from Sigma-Aldrich (Saint Louis, MO, USA). Polyethylenimine (PEI) was from Polysciences (Warrington, PA, USA) and protein-G-Sepharose and glutathione-Sepharose were from GE-Healthcare (Little Chalfont, Buckinghamshire, England). [γ-32P]-ATP was from Perkim-Elmer (Waltham, MA, USA).

### 4.2. Antibodies

The polyclonal antibodies anti-ERK5 and anti-phospho ERK5-pT^218^EY^220^ were from Cell Signaling Technology (Danvers, MA, USA). The anti-ERK5 antibody used in immunoprecipitation and immunofluorescence experiments (C-7) and the anti-Cdc37 and anti-CREB1 antibodies were from Santa Cruz Biotechnology (Dallas, TX, USA). The anti-GST, anti-FLAG and anti-HA monoclonal antibodies were from Sigma-Aldrich (Saint Louis, MO, USA); the anti-Hsp90β from Merck Merck Millipore (Burlington, MASS, USA); and the anti-SUMO antibody from Invitrogen (Carlsbad, CA, USA). The anti-Ubc9 antibody was a gift of Dr. R. Hay (College of Life Sciences, University of Dundee, Dundee, UK).

### 4.3. DNA Constructs

Recombinant DNA procedures were performed using standard protocols. The pEBG-2T vectors encoding for GST-tagged human ERK5 full length (WT), kinase inactive mutant (D200A) or N-terminus (aa 1-490) were a gift from Dr. P. Cohen (MRC Protein Phosphorylation Unit, Dundee, UK) [51], and they have been described before [29]. The vector containing human ERK5 SUMO-deficient K6/22R was from Dr. J-I Abe (University of Rochester, NY), and it was used to generate N-term GST-tagged ERK5 K6/22R mutant in a pEBG-2T vector. The pCMV plasmid encoding HA-tagged MEK5-DD (constitutively active) was from Dr. E. Nishida (Kyoto University, Japan, [27]), the pCDNA3 vector encoding FLAG-tagged Cdc37 was a gift of Dr. Scheidereit (Center for Molecular Medicine, Berlin, Germany [52]) and the pcDNA3.1 vector encoding HA-tagged Hsp90β was from Dr. Papapetropoulos (University of Athens, Greece, [53]). pCDNA3 vectors encoding Ubc9 and 6xHis-tagged SUMO2 were from Dr. R. Hay (College of Life Sciences, University of Dundee, Dundee, UK). AP1-luciferase vector was purchased from Stratagene (La Jolla, CA, USA), and pRL-CMV-Renilla was from Promega (Madison, WI, USA).

### 4.4. Cell Culture, Transfection and Lysis

Cells were cultured at 37 °C under humidified air (5% CO_2_). Human HEK293T, HeLa and PC-3 cells were cultured in Dulbecco’s modified Eagle’s medium (DMEM) supplemented with 10% FBS and antibiotics. Cells were transfected using polyethylenimine, as described before [54]. Unless otherwise stated, cells were lysed in ice-cold lysis buffer (50 mM Tris-HCl, pH 7.5, 1 mM EGTA, 1 mM EDTA, 1% (*w*/*v*) NP-40, 1 mM sodium orthovanadate, 10 mM sodium-β-glycerophosphate, 50 mM sodium fluoride, 5 mM sodium pyrophosphate, 0.27 M sucrose, 0.1% (vol/vol) 2-mercaptoethanol and complete protease inhibitor cocktail). Lysates were centrifuged at 12,000 *g* for 12 min at 4 °C, and supernatants were stored at −20 °C. Protein concentration was determined by the Bradford method [55].

### 4.5. Immunofluorescence Microscopy

Cells grown on polylysine-coated cover slips were fixed with 4% paraformaldehyde for 20 min and mounted in medium with Hoechst 33259 for DNA staining. To monitor endogenous ERK5, fixed cells were processed as described before [29] using anti-ERK5 antibody (C-term, Santa Cruz Biotechnology, Dallas, TX, USA) and the corresponding fluorescent-labeled secondary antibody and Hoechst 33259. Cells were visualized by fluorescence microscopy in a Nikon Eclipse 90i epifluorescence microscope (Tokyo, Japan).

### 4.6. Immunoprecipitation and Immunoblotting

Protein G-sepharose beads (10 μL) bound to 2 μg of the corresponding antibody were incubated with 0.5 mg of cell lysate for 2 h at 4 °C. The immunoprecipitates were washed twice with lysis buffer containing 0.15 M NaCl, twice with buffer A (50 mM Tris-HCl, pH 7.5, 0.1 mM EGTA and 0.1% 2-mercaptoethanol) and the immune complexes eluted in 2× sample buffer. GST-, HA- and FLAG-tagged overexpressed proteins were immunoprecipitated as described above, using the appropriate resin. Immunoblotting was performed as described previously [56].

### 4.7. In Vivo ERK5 SUMOylation Assay

We followed the method described by Tatham et al. [57]. HEK293T cells overexpressing His-tagged SUMO2, the E2 ligase Ubc9 and the indicated form of ERK5 were resuspended in 5 mL of buffer 1 (6 M guanidinium–HCl, 10 mM Tris100 mM and 100 mM Na2HPO4/NaH2PO4 buffer, pH 8); sonicated for 1 min and centrifuged at 5000 *g* for 5 min. The resulting supernatants were incubated with 20 µL of Ni2+-NTA-agarose beads for 2 h at room temperature with rotation. Beads were successively washed as follows: twice with 4 mL of Buffer 1 plus 10 mM 2-mercaptoethanol; three times with 4 mL of buffer 2 (8 M urea, 10 mM Tris, 10 mM 2-mercaptoethanol and 100 mM Na2HPO4/NaH2PO4 buffer, pH 8); twice with 4 mL of buffer 3 (8 M Urea, 10 mM Tris and 100 mM Na2HPO4/NaH2PO4 buffer, pH 6.3) containing 10 mM 2-mercaptoethanol; once with 1 mL of buffer 3 containing 0.2% Triton X-100; once with 1 mL of buffer 3 containing 0.1% Triton X-100 and 0.5 M NaCl and three washes with 1 mL of buffer 3. Finally, proteins were eluted by incubating the beads with 200 mM imidazole in 5% SDS, 0.15 M Tris–HCl, pH 6.7, 30% (*v*/*v*) glycerol and 0.72 M 2-mercaptoethanol for 1 h at 37 °C with mixing.

In parallel experiments, HEK293T cells overexpressing His-tagged SUMO2, the E2 ligase Ubc9 and the indicated FLAG-tagged ERK5, alone or in combination with MEK5DD or Cdc37, were resuspended in 1 mL of RIPA buffer (25 mM Tris-HCl, pH 7.9, 150 mM NaCl, 1 mM EGTA, 5 mM pirofosfato sódico, 0.5% (*w*/*v*) ácido desoxicólico, 0.1% (*w*/*v*) SDS and 1% (*w*/*v*) NP-40) containing 10 mM NEM. After sonication, cell extracts were incubated with 2 μg anti-ERK5 antibody bound to protein G-sepharose beads for 2 h at 4 °C. After washing the beads three times with RIPA buffer containing 0.5 M NaCl and one time with wash buffer (50 mM Tris-Cl, pH 7.5 and 0.1 mM EGTA), bound ERK5 was released in 5× Laemli sample buffer.

### 4.8. Subcellular Fractionation

PC-3 cells overexpressing the indicated proteins were resuspended in 600 μL of fractionation buffer (10 mM HEPES, pH 7.4, 250 mM sucrose, 1 mM EGTA, 1 mM orthovanadate, 50 mM NaF and 5 mM sodium pyrophosphate) and kept on ice for 10 min. All the procedures were performed at 4 °C. Cells were mechanically lysed in a potter homogenizer equipped with a teflon pestle (Braun, 40 strokes) and centrifuged at 2000 *g* for 15 min. Supernatants containing the cytosolic fraction were collected and kept on ice. Pellets containing the nuclear fraction were washed once with fractionation buffer, and the resulting pellets were resuspended in 300 μL of RIPA buffer, passed through a 25-gauge needle (Braun) using a 1-mL syringe (Braun), kept on ice for 10 min and centrifuged for 12,000 *g* for 12 min. Supernatants containing the nuclear fraction were stored at −20 °C.

### 4.9. ERK5 kinase Activity Assay

HEK293T cell extracts (0.5 mg) overexpressing different forms of GST-tagged ERK5 and the indicated proteins were incubated for 1 h at 4 °C with 10 μL of glutathione-sepharose beads. Beads were then washed twice with lysis buffer containing 0.5 M NaCl, followed by two washes with buffer A. Kinase activity assay was performed in a assay volume of 50 μL containing glutathione-sepharose beads, buffer A, 10 mM magnesium acetate and 0.1 mM [γ-32P]-ATP (500 cpm/pmol) and 500 μM PIMtide (ARKKRRHPSGPPTA), a bona fide ERK5 peptide substrate [29]. Assays were carried out for 45 min at 30 °C, terminated by applying the reaction mixture onto p81 paper, and the incorporated radioactivity was measured as described previously [29]. One milli-unit of activity is the amount of enzyme that catalyzes the phosphorylation of 1 pmol of PIMtide in 1 min.

### 4.10. Reporter Luciferase Assay

Cells cultured in 12-wells plates were transfected with 650 ng of DNA, which contained 100 ng of AP-1-driven luciferase reporter construct and 50 ng Renilla and the indicated amounts of plasmids. Twenty-four hours later, luciferase activity assay was performed using the Dual-Luciferase kit (Promega, Madison, WI, USA), following the manufacturer’s instructions.

### 4.11. Statistical Analysis

Figures were generated using Microsoft PowerPoint 15.15 (Redmond, WA, USA) or Adobe Photoshop software 7.0 (San José, CA, USA). Statistical significance was determined using Prism 4.0 software (San Diego, CA, USA), using two-way ANOVA followed by a Tukey’s test.

## Figures and Tables

**Figure 1 ijms-21-02203-f001:**
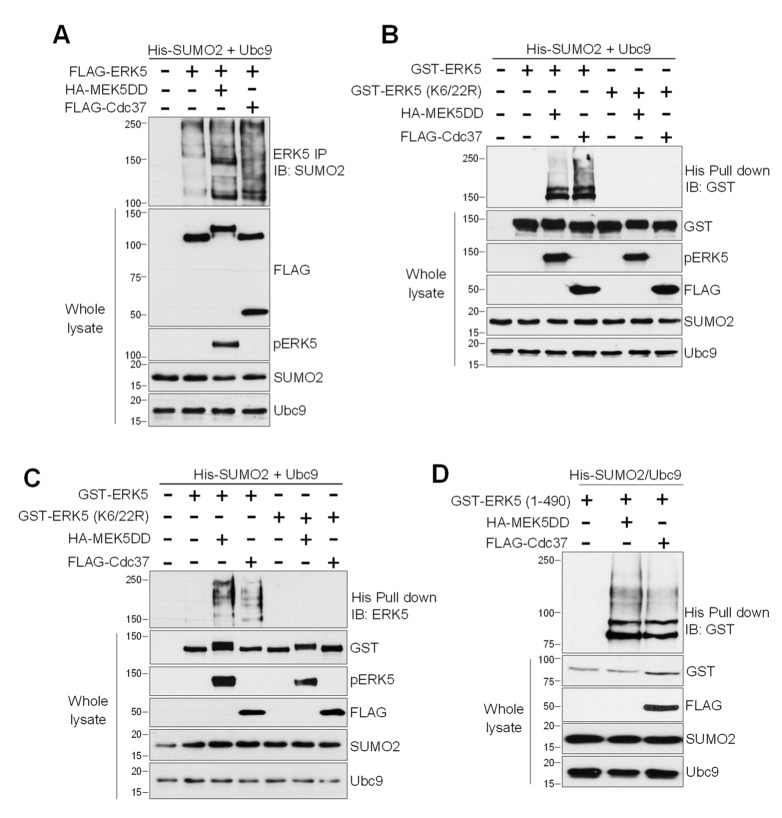
MEK5 phosphorylation or Cdc37 overexpression induce ERK5 Small Ubiquitin-related Modifier (SUMO)ylation. HEK293T cells we co-transfected with His-tagged SUMO2 and Ubc9 and the indicated combinations of FLAG-tagged ERK5 (**A**) or GST-tagged ERK5 (wild type or SUMO-deficient K6/22R mutant) (**B**,**C**). Cells were also transfected with MEK5DD (constitutively active) or Cdc37 to induce ERK5 nuclear translocation. After lysing cells with RIPA buffer containing NEM, ERK5 was immunoprecipitated, and SUMOylated species were detected by immunoblotting with anti-SUMO2 antibody (**A**). In parallel experiments, cells were lysed with denaturing buffer containing NEM, and SUMOylated ERK5 was affinity-purified using Ni^2+^-agarose beads and detected by immunoblotting with anti-GST (**B**) or anti-ERK5 (**C**) antibody. Levels of overexpressed proteins and phosphorylated ERK5 at the T-loop are shown using the corresponding antibodies. (**D**)SUMOylation analysis of ERK5 N-terminal half. Experiments were performed as in (**B**), using a mutant form of ERK5 that encodes for the aa 1–490 which contains the kinase domain (aa 54–346). Similar results were obtained in three separate experiments.

**Figure 2 ijms-21-02203-f002:**
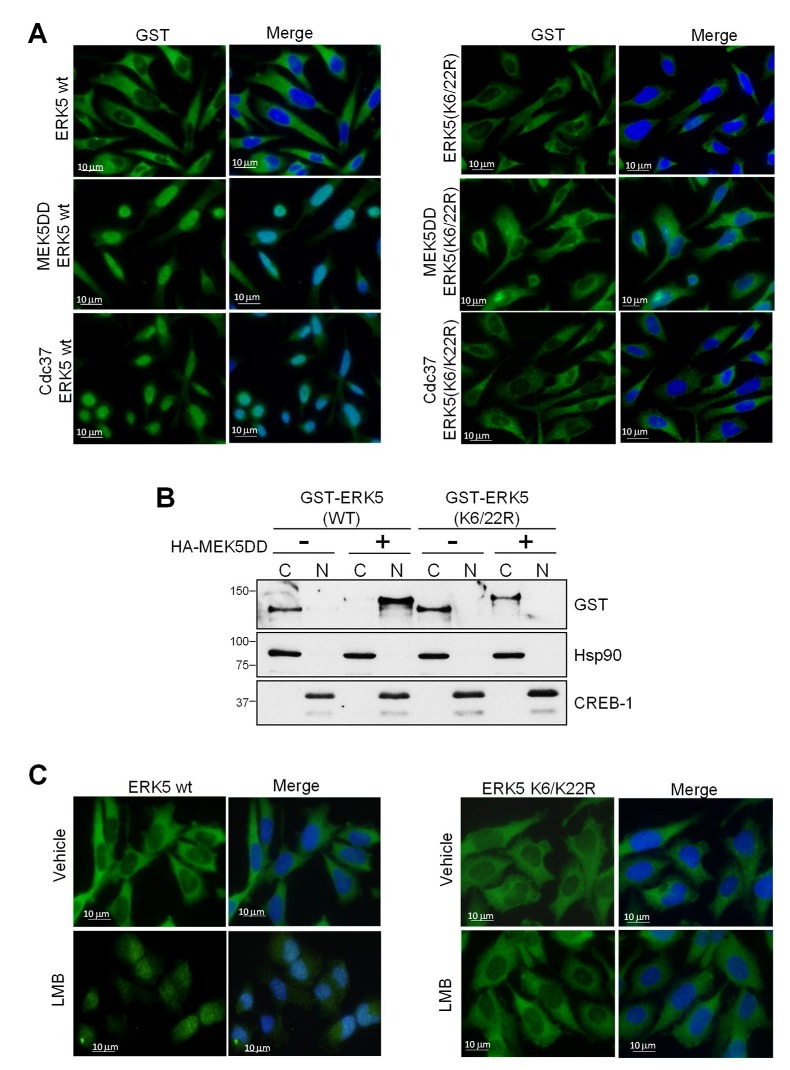
ERK5 requires SUMO modification for nuclear translocation. Immunofluorescence microscopy. (**A**) PC-3 cells were transfected with either GST-tagged ERK5 (wild type or K6/22R SUMO-deficient mutant) and MEK5DD or Cdc37 plasmids. After 24 h, cells were fixed with paraformaldehyde and immunofluorescent staining for ERK5 (green) using anti-GST antibody. Nuclei were stained with Hoechst (blue). (**B**) Subcellular fractionation. PC-3 cells were transfected with the indicated vectors and submitted to cellular fractionation as described in the Materials and Methods section. Proteins were detected by immunoblot using the indicated antibodies. (**C**) PC-3 cells were transfected with either GST-tagged ERK5 wild type (WT) or K6/22 mutant, left alone or treated with the nuclear export inhibitor leptomycin B (LMB, 20 nM) for 18 h and stained as in A. Similar results were obtained in three independent experiments. Scale bars, 10 μm.

**Figure 3 ijms-21-02203-f003:**
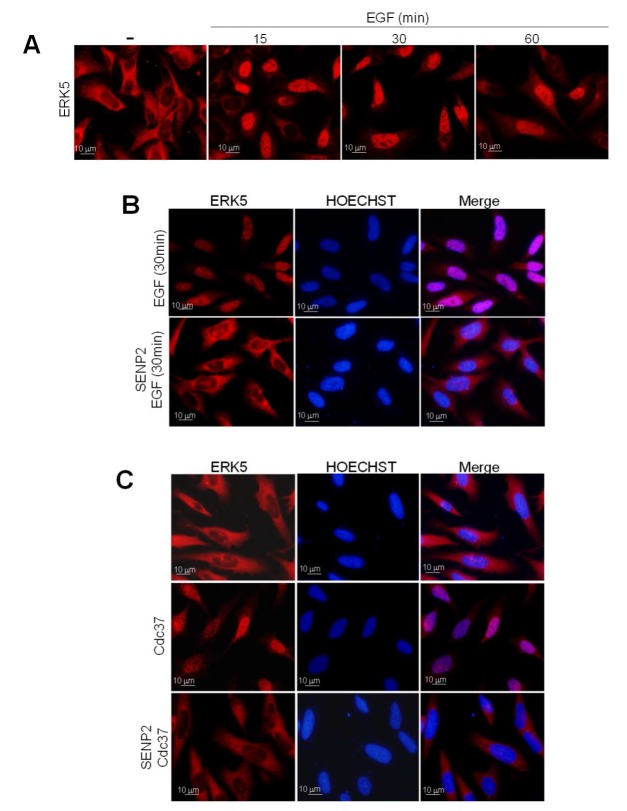
Overexpression of SUMO peptidase SENP2 inhibits nuclear translocation of endogenous ERK5. (**A**) ERK5 nuclear translocation in response to epidermal growth factor (EGF). PC-3 cells were starved prior to stimulation with 50-ng/mL EGF. At the indicated times, cells were fixed and immunofluorescent stained for ERK5. (**B**) Effect of SENP2 on EGF-mediated ERK5 nuclear translocation. Transfected cells with empty vector (control) or vector encoding for SUMO protease SENP2. Thirty-six h later, cells were starved for 16 h prior to stimulation with 50 ng/mL for 30 min. Cells were immunostained for endogenous ERK5 (red) and nuclei (Hoechst, blue). (**C**) Effect of SENP2 on Cdc37-mediated ERK5 nuclear translocation. Transfected cells with empty vector (control) or vector encoding for Cdc37 alone or in combination with a vector encoding SENP2 were fixed, and ERK5 was visualized by immunostaining (red). Similar results were obtained in three independent experiments. Scale bars, 10 μm.

**Figure 4 ijms-21-02203-f004:**
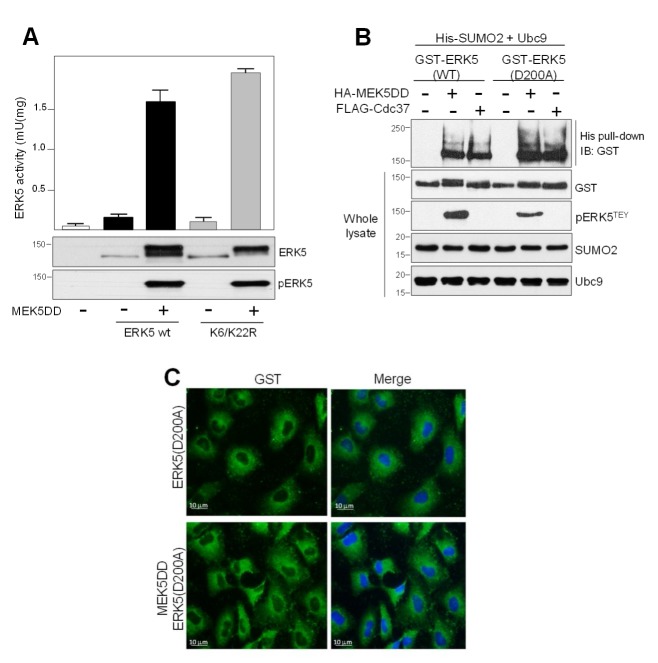
ERK5 SUMOylation is not necessary for ERK5 kinase activity. (**A**) HEK293T cells overexpressing GST-tagged ERK5 (wild type or SUMO-deficient K6/22R mutant), alone or in combination with MEK5DD, were lysed, and the GST-ERK5 protein was affinity-purified and assayed for kinase activity, as described in the Methods Section. Activity data are the mean ± SD of two independent experiments, each performed in triplicate. Cell lysates were also immunoblotted for ERK5 (anti-GST antibody) and for phosphorylated ERK5 (pERK5). (**B**) HEK293T cells were co-transfected with His-tagged SUMO2 and Ubc9, and the indicated combinations of GST-tagged ERK5-D200A (kinase inactive), HA-tagged MEK5DD (constitutively active) and FLAG-tagged Cdc37. After lysing the cells with denaturing buffer containing NEM, SUMOylated ERK5 was affinity-purified using Ni^2+^-agarose beads and detected by immunoblotting with anti-GST antibody. Levels of overexpressed proteins and phosphorylated ERK5 at the T-loop are shown using the corresponding antibodies. Similar results were obtained in three independent experiments. (**C**) Immunofluorescence microscopy. PC-3 cells were transfected with GST-tagged ERK5 (wild type or kinase-inactive mutant D200A) and MEK5DD plasmids. After 24 h, cells were fixed with paraformaldehyde and immunofluorescent staining for ERK5 (green) using anti-GST antibody. Nuclei were stained with Hoechst (blue).

**Figure 5 ijms-21-02203-f005:**
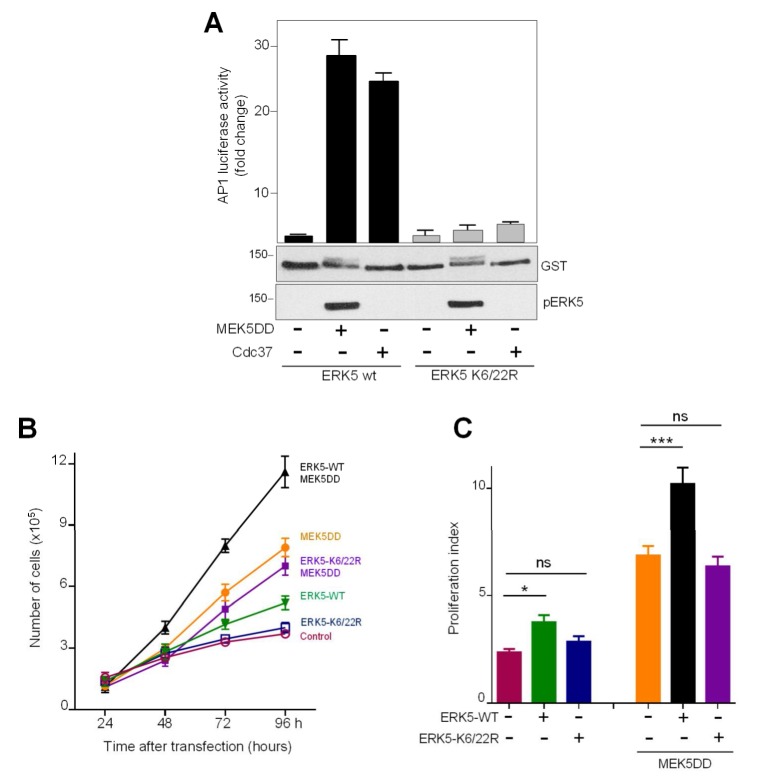
ERK5 SUMOylation is necessary for transcriptional activity and cell proliferation. (**A**) SUMO modification is required for ERK5-mediated AP-1 transcriptional activity. pAP-1 luciferase reporter and pRL-CMV-Renilla plasmids were co-transfected with the indicated plasmids in HeLa cells. Twenty-four hours later, lysates were subjected to a dual-luciferase assay. Each value is the mean ± SD of three separate determinations, each performed in triplicate and normalized using the Renilla values. Cell lysates were also immunoblotted for ERK5 (anti-GST antibody) and for phosphorylated ERK5 (pERK5). (**B**) ERK5 SUMOylation is necessary for ERK5-mediated cell proliferation. PC-3 cells were transfected with plasmids encoding for ERK5-WT, ERK5-K6/22 mutant, MEK5DD or for ERK5-WT and MEK5DD or ERK5-K6/22R and MEK5DD. At the indicated times, cells were counted as described in Materials and Methods. (**C**) Shows the corresponding proliferation curves; right panel shows the corresponding proliferation index values, measured as the ratio number cells at day 4:number of cells at day 1. Each value is the mean ± SD of the results determined for three different transfected cell dishes. Similar results were obtained in two independent experiments. ns: not significant; * *p* < 0.05 and *** *p* < 0.001.

**Figure 6 ijms-21-02203-f006:**
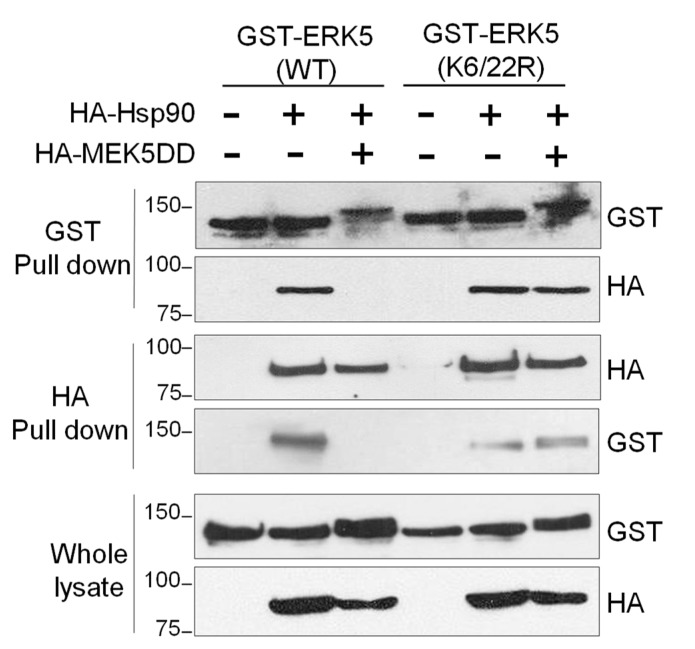
SUMOylation is necessary for Hsp90 dissociation from the ERK5-Cdc37 complex after MEK5 activation. HEK293T cells were transfected with GST-tagged ERK5 (wild type or SUMO-deficient K6/22R mutant) alone or in combination with Hsp90 and MEK5DD. Thirty-six hours later, cells were lysed, and GST-ERK5 or HA-Hsp90 proteins were affinity-purified using glutathione-sepharose or anti-HA agarose resins, respectively. Immune complexes were immunoblotted for ERK5 and Hsp90. Similar results were obtained in three independent experiments.

**Figure 7 ijms-21-02203-f007:**
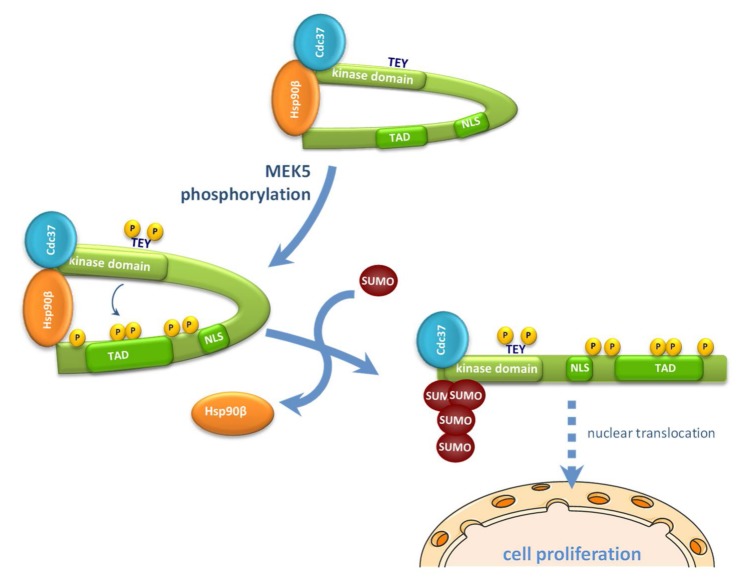
A model for the role of SUMO modification in the mechanism of nucleocytoplasmic shuttling of ERK5. NLS, nuclear localization signal; TAD, transactivation domain; TEY, Thr218/Tyr220.

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
