# Peer review of "SUMOylation Is Required for ERK5 Nuclear Translocation and ERK5-Mediated Cancer Cell Proliferation"

_ijms, 2020, doi:10.3390/ijms21062203_

Round 1
Reviewer 1 Report
In this well written manuscript, Tatiana Erazo and colleagues definitely demonstrate that ERK5 is SUMOylated at residues Lys6/Lys22 in cancer cells upon constitutively active MEK5 or cdc37 overexpression. More importantly they show that this modification positively regulates the amount and activity (as a transcriptional trans regulator) of ERK5 in the nucleus, as well as PC3 proliferation. The experiments are well conducted and support the majority of the driven conclusions. Some minor points should be addressed in order to definitely substantiate some issues.
Why in Figure 1a in the membrane immunoblotted with the anti GST Ab performed on whole cell lysates, there is only a shifted band in the presence of MEK5DD (lanes 3 and 6)? Is this shift due to ERK5 phosphorylation only, while slower migration following SUMOylation is not evident? Please, give an explanation for that. Moreover, why is pERK5 (lane 3) of the same MW of pERK5 in lane 6? I was expecting a higher MW of the first with respect to the second due to SUMOylation.
Is ERK5 SUMOylation appreciable without overexpression? Does the pretreatment with SENP2 inhibitors, or any other approach, allow to evidentiate “physiologic” ERK5 SUMOYlation (i.e. without overexpression) and increased ERK5 nuclear localization in cancer cells? Indeed, Figure 3 shows the modulation of nuclear shuttling of endogenous ERK5 but it is not evident that this occurs upon SUMOylation (co-localization of SUMO and ERK5 is not showed).
Concerning the experimental setting: it might be useful to perform a western blot assay with nuclear fraction and cytoplasmic fraction lysates of, for example, SUMO-deficient mutants or of other conditions tested in the work.
Figure 1a and b: MW should be indicated in all wb. I would like to see antiERK5 immunoblotting following the pull-down assay, at least in Figure 1a. Is the amount of SUMOylated ERK5 too small to be evident? Is it possible that the low amount does not allow to see slower migrating bands in the anti GST immunoblotted membranes? Along the same line, I would like to see immunoprecipitated ERK5 immunobloted for SUMO2.
Page 6, lines 11-13: please discuss these results in light of those previously published by the same authors (Mol. Cell. Biol. 2013, 33, 1671–1686.) indicating that overexpression of cdc37 induces translocation of wild type ERK5 as well as of the kinase-inactive form (D200A).
Figure 4 b. It seems that D200A ERK5 is more sumoylated than ERK5 wt. Is this the case?
Figre 4b. Characters, Letterds in the garpg are too small
Few typos and minor points are listed below
Abstract:
Line 18: MEK5 or overexpression of cdc37
Line 19: mechanisms that induce nuclear ERK5 should read: mechanisms that increase nuclear ERK5/ mechanisms that induce nuclear translocation of ERK5
The sentence “We also show that overexpression of the SUMO protease SENP2 completely abolished endogenous ERK5 nuclear localization in response to EGF stimulation.” should be located following “Furthermore, mutation of these SUMO sites abolished the ability of ERK5 to translocate to the nucleus and to promote prostatic cancer PC-3 cell proliferation.”
Page 1 , line 33: contains a N-terminal kinase domain should read : contains an N-terminal kinase domain
Page 1, line 31: ERK5 misses round bracket;
Page 1, line 36: “through” instead of “though”;
Page 2, line 3: remove round brackets from the two citations;
Page 2, lines 20-21: choose whether to put the names of amino acids as a symbol or as a one-letter symbol to standardise the text;
Page 4, line 4 or line 14: It might be useful to indicate the concentration of Leptomycin B used in the experiment;
Page 4, line 25: blank space missed between “endogenous” and “ERK5”
Page 8, line 28: replace “thirty-six hour” withs 36h
I would cite Front Cell Dev Biol 2016, 4: 105, in the introduction, in order to further underline the relevance of ERk5 nuclear localization in cancer cell proliferation. Along the same line, the recent article from Tubita et al (Int J Mol Sci. 2020 Jan 31;21(3)) would additionally support the relevance of the findings described in the present paper.
Author Response
We would like to thank you for reviewing our manuscript. We found the criticisms fair and constructive. This has prompted us to perform the experiments requested, resulting in strengthened revised manuscript.
Specifically, we have performed new experiments, which are itemized below:
- New Figure 1. New figure 1 contains now two new panels, showing the results after immunoprecipitating ERK5 protein and staining with anti-SUMO2 antibody (Figure 1A), as well as those ones after pulling-down SUMO2 species with Ni2+-agarose beads and detected with anti-ERK5 antibody (Figure 1C).
- New Figure 2. The new Figure 2 contains a new panel showing ERK5 SUMOylation results after a subcellular fractionation of PC-3 cells overexpressing either ERK5 WT or the SUMO-deficient mutant.
According to those experiments and to reviewers concerns, the text has been revised. We have made obvious the new text and data of the new version of the paper, by marking in red all the changes.
Here we report our itemized response to the criticisms:
In this well written manuscript, Tatiana Erazo and colleagues definitely demonstrate that ERK5 is SUMOylated at residues Lys6/Lys22 in cancer cells upon constitutively active MEK5 or cdc37 overexpression. More importantly they show that this modification positively regulates the amount and activity (as a transcriptional trans regulator) of ERK5 in the nucleus, as well as PC3 proliferation. The experiments are well conducted and support the majority of the driven conclusions. Some minor points should be addressed in order to definitely substantiate some issues.
Why in Figure 1a in the membrane immunoblotted with the anti GST Ab performed on whole cell lysates, there is only a shifted band in the presence of MEK5DD (lanes 3 and 6)?
Is this shift due to ERK5 phosphorylation only, while slower migration following SUMOylation is not evident? Please, give an explanation for that. Moreover, why is pERK5 (lane 3) of the same MW of pERK5 in lane 6? I was expecting a higher MW of the first with respect to the second due to SUMOylation.
SUMOylation is very dynamic process, and the amount of SUMOylated proteins in a giving time is barely detected. This is more obvious for the SUMO2/3 forms, since they also suffer branched SUMOylation and, therefore, the amount of ERK5 SUMOylated species (containing different lengths of SUMO branches) is low. For our experiments, we use 10 g of cell lysates to perform the immunoblot analyses for overexpressed proteins and phosphorylated ERK5. However, we had to pull-down or immunoprecipitate 3 mg of protein lysates, as well as to use highly concentrated primary antibodies, in order to detect the SUMOylated forms of ERK5. Therefore, the shift observed in ERK5 immunoblots in response to MEK5 overexpression is due to the canonical shift observed for active ERK5 (since SUMOylated forms cannot be detected when analysing 10 g of lysates).
Is ERK5 SUMOylation appreciable without overexpression? Does the pretreatment with SENP2 inhibitors, or any other approach, allow to evidentiate “physiologic” ERK5 SUMOYlation (i.e. without overexpression) and increased ERK5 nuclear localization in cancer cells? Indeed, Figure 3 shows the modulation of nuclear shuttling of endogenous ERK5 but it is not evident that this occurs upon SUMOylation (co-localization of SUMO and ERK5 is not showed).
For most SUMO substrates, only a low proportion of the cellular pool is modified by SUMO. Thus, the detection of a given SUMOylated protein without overexpression of SUMO or the substrate is technically very challenging. We have tried to perform experiments with endogenous ERK5 but did not rendered clear results. However, other authors have convincingly demonstrated ERK5 SUMOylation (Woo et al., Circ. Res. 102: 538-45, 2008; Heo et al., Circ. Res. 112: 911-23, 2013). The purpose of our work was to investigate the role of SUMOylation in the nuclear shuttling of ERK5. In this regard, we believe that we convincingly show that the ERK5 SUMO-deficient mutant does not translocate to the nucleus under no circumstance. Furthermore, given the fact that EGF induces MEK5-mediated activation of ERK5, and that active MEK5 induces ERK5 SUMOylation, the experiments performed with endogenous ERK5 show that canonical ERK5 nuclear translocation is driven by SUMO modification, since overexpression of ERK5 SUMO protease SENP2 effectively blocked this process. Finally, there are no specific SUMO2 inhibitors available, but rather unspecific SUMO1/2/3 inhibitors that will make difficult to draw conclusions (SUMO1 is implicated in many cellular processes).
Concerning the experimental setting: it might be useful to perform a western blot assay with nuclear fraction and cytoplasmic fraction lysates of, for example, SUMO-deficient mutants or of other conditions tested in the work.
As requested, we have performed subcellular fractionation assay in PC-3 cells overexpressing either ERK5 WT or the SUMO-deficient mutant. The new Figure 2B shows the expected nuclear translocation of wild type ERK5 in response to MEK5DD overexpression. On the contrary, SUMO-deficient mutant (K6/22R) did not translocate to nucleus after MEK5DD overexpression, in spite of this mutant showing the expected shifted band corresponding to activated ERK5 (also shown in Figure 4A). These results confirm those obtained in IF experiments (Figure 2A) and, overall, they show that SUMO-deficient ERK5 does not translocate to nucleus after MEK5-mediated activation.
Figure 1a and b: MW should be indicated in all wb. I would like to see antiERK5 immunoblotting following the pull-down assay, at least in Figure 1a. Is the amount of SUMOylated ERK5 too small to be evident? Is it possible that the low amount does not allow to see slower migrating bands in the anti GST immunoblotted membranes? Along the same line, I would like to see immunoprecipitated ERK5 immunobloted for SUMO2.
As requested, all the MW are now indicated in each of the figures containing immunoblots.
We have performed the requested experiments. New Figure 1A shows the results after immunoprecipitating ERK5 protein and staining with anti-SUMO2 antibody. As it happens for the experiments in which all SUMO2 species were pulled-down, we observed an increase on the levels of the slow migrating species of ERK5 in response to either MEK5DD or Cdc37 overexpression.
The new Figure 1C shows the results after pulling-down SUMO2 species with Ni2+-agarose beads and detected with anti-ERK5 antibody. As expected, we obtained analogous results to those obtained when SUMOylated ERK5 was detected using the anti-GST antibody.
Page 6, lines 11-13: please discuss these results in light of those previously published by the same authors (Mol. Cell. Biol. 2013, 33, 1671–1686.) indicating that overexpression of cdc37 induces translocation of wild type ERK5 as well as of the kinase-inactive form (D200A).
The following text have been included in the Discussion section of the new version of the manuscript:
“We previously described that overexpression of Cdc37 also results in Hsp90 dissociation and ERK5 nuclear shuttling. This mechanism does not require ERK5 kinase activity, since Cdc37 overexpression induces nuclear translocation of the kinase dead mutant D200A, or of the wild type ERK5 even in the presence of XMD8-92 inhibitor [29]. Here, we show that Cdc37 overexpression results in SUMOylation of the ERK5-D200A mutant (Figure 4B). Thus, it is likely that binding of the overexpressed Cdc37 would induce a conformational change in ERK5, independently of its kinase activity, and similar to that induced by the autophosphorylation of the C-terminal tail. This change, then, would facilitate the recruitment of the putative SUMO ligase, resulting in SUMOylation and the nuclear shuttling of ERK5. Further experiments will be required in order to identify the domain of ERK5 involved in the binding of the SUMO ligase, and to shed light into the precise mechanism involved in ERK5 nuclear translocation.”
Figure 4 b. It seems that D200A ERK5 is more sumoylated than ERK5 wt. Is this the case?
This is an interesting question. In spite of that quantification of SUMO species is technically difficult, we have observed (three experiments) slightly higher SUMO signal for the D200A mutant. We have no explanation, but we could speculate that phosphorylation of the C-term tail might be involved in a downregulation mechanism, partially facilitating the interaction with the SENP2 protease
Figure 4b. Characters, Letters in the graph are too small
As requested, text in Figure 4B has been enlarged
Few typos and minor points are listed below
Abstract:
Line 18: MEK5 or overexpression of cdc37
Line 19: mechanisms that induce nuclear ERK5 should read: mechanisms that increase nuclear ERK5/ mechanisms that induce nuclear translocation of ERK5
The sentence “We also show that overexpression of the SUMO protease SENP2 completely abolished endogenous ERK5 nuclear localization in response to EGF stimulation.” should be located following “Furthermore, mutation of these SUMO sites abolished the ability of ERK5 to translocate to the nucleus and to promote prostatic cancer PC-3 cell proliferation.”
Page 1 , line 33: contains a N-terminal kinase domain should read : contains an N-terminal kinase domain
Page 1, line 31: ERK5 misses round bracket;
Page 1, line 36: “through” instead of “though”;
Page 2, line 3: remove round brackets from the two citations;
Page 2, lines 20-21: choose whether to put the names of amino acids as a symbol or as a one-letter symbol to standardise the text;
Page 4, line 4 or line 14: It might be useful to indicate the concentration of Leptomycin B used in the experiment;
Page 4, line 25: blank space missed between “endogenous” and “ERK5”
Page 8, line 28: replace “thirty-six hour” withs 36h
The new version of the manuscript contains now the requested changes, which are marked in red. Regarding the letter/code for amino acids, we have used the three letters code for the text, and the one letter code for the Figures.
I would cite Front Cell Dev Biol 2016, 4: 105, in the introduction, in order to further underline the relevance of ERk5 nuclear localization in cancer cell proliferation. Along the same line, the recent article from Tubita et al (Int J Mol Sci. 2020 Jan 31;21(3)) would additionally support the relevance of the findings described in the present paper.
The suggested paper is now cited in the new version of the manuscript. This paper has been cited at the discussion section, where the results about ERK5-mediated proliferation are discussed.
Reviewer 2 Report
In this MS Erazo et al, address the role of SUMOylation in ERK5 nuclear translocation. They previously described that the molecular chaperone Hsp90 stabilizes and anchors ERK5 at the cytosol, and that ERK5 nuclear shuttling requires Hsp90 dissociation. Here, they show that MEK5 or Cdc37 overexpression induce ERK5 SUMO-2 modification at residues Lys6/Lys22. They also show that overexpression of the SUMO protease SENP2 completely abolished endogenous ERK5 nuclear localization in response to EGF stimulation. Mutation of these SUMO sites abolished the ability of ERK5 to translocate to the nucleus and to promote prostatic cancer PC-3 cell proliferation without affecting ERK5 kinase activity.
The novelty of this work is mild because it has been published that SUMOylation of these residues controls ERK5 transcriptional activity. The results are clear and experiments have the correct controls, but physiological effects are meanly described after overexpression of proteins or constitutively active mutants. The experiment that shows that endogenous ERK5 requires SUMOylation for shuttling is based on SENP overexpression that could affect multiple proteins. The ERK5 Lys6/Lys22 mutant can have a conformational change that precludes its correct translocation and transactivation activity. Authors should show in Fig. 2 that decreasing SUMO2 by shRNA decreases EGF-induced translocation. Conversely, shRNA SENP should increase translocation.
The same approach should be done in Fig. 5 experiments to show that SUMOylation is effectively required for ERK5 transcriptional activity and proliferation.
Author Response
We would like to thank you for reviewing our manuscript. We found the criticisms fair and constructive. This has prompted us to perform the experiments requested, resulting in strengthened revised manuscript.
Specifically, we have performed new experiments, which are itemized below:
- New Figure 1. New figure 1 contains now two new panels, showing the results after immunoprecipitating ERK5 protein and staining with anti-SUMO2 antibody (Figure 1A), as well as those ones after pulling-down SUMO2 species with Ni2+-agarose beads and detected with anti-ERK5 antibody (Figure 1C).
- 2. New Figure 2. The new Figure 2 contains a new panel showing ERK5 SUMOylation results after a subcellular fractionation of PC-3 cells overexpressing either ERK5 WT or the SUMO-deficient mutant.
According to those experiments and to reviewers concerns, the text has been revised. We have made obvious the new text and data of the new version of the paper, by marking in red all the changes.
Here we report our itemized response to the criticisms:
In this MS Erazo et al, address the role of SUMOylation in ERK5 nuclear translocation. They previously described that the molecular chaperone Hsp90 stabilizes and anchors ERK5 at the cytosol, and that ERK5 nuclear shuttling requires Hsp90 dissociation. Here, they show that MEK5 or Cdc37 overexpression induce ERK5 SUMO-2 modification at residues Lys6/Lys22. They also show that overexpression of the SUMO protease SENP2 completely abolished endogenous ERK5 nuclear localization in response to EGF stimulation. Mutation of these SUMO sites abolished the ability of ERK5 to translocate to the nucleus and to promote prostatic cancer PC-3 cell proliferation without affecting ERK5 kinase activity.
The novelty of this work is mild because it has been published that SUMOylation of these residues controls ERK5 transcriptional activity. The results are clear and experiments have the correct controls, but physiological effects are meanly described after overexpression of proteins or constitutively active mutants. The experiment that shows that endogenous ERK5 requires SUMOylation for shuttling is based on SENP overexpression that could affect multiple proteins. The ERK5 Lys6/Lys22 mutant can have a conformational change that precludes its correct translocation and transactivation activity.
The reviewer states that “the novelty of this work is mild because it has been published that SUMOylation of these residues controls ERK5 transcriptional activity”. We disagree given that this study is the first to reveal SUMO modification as a postraslational mechanism necessary for ERK5 shuttling to the nucleus. Moreover, we have integrated this into a more detailed mechanism that describes the interplay between two different posttranslational modifications (ERK5 phosphorylation and SUMO modification) and the regulation of ERK5 by chaperones.
SUMOylation is very dynamic process, and the amount of SUMOylated proteins in a giving time is barely detected. This is more obvious for the SUMO2/3 forms, since they also suffer branched SUMOylation and, therefore, the amount of ERK5 SUMOylated species (containing different lengths of SUMO branches) is low. Because of that, for most SUMO substrates only a low proportion of the cellular pool is modified by SUMO. Thus, the detection of a given SUMOylated protein without overexpression of SUMO or the substrate is technically very challenging. Abe’s group demonstrated that ERK5 becomes SUMOylated at Lys6 and Lys22. Therefore, we have used the mutant K6/22R to perform our work. The use of SUMO-deficient mutants is widely accepted to investigate the role of this postranslational modification on a target protein. In our case, we clearly show that this mutant is functional, given: a) is phosphorylated and activated by MEK5 (Figures 1B, 1C, 4A and 5A); b), is an stable protein, since the expression level are similar to the wild type form (Figures 1B, 1C, 4A, 5A and 6); and c), binds to Hsp90 as it does the WT protein. Considering those, it is likely that SUMO modification might well preclude the conformation change required for ERK5 nuclear translocation.
Authors should show in Fig. 2 that decreasing SUMO2 by shRNA decreases EGF-induced translocation. Conversely, shRNA SENP should increase translocation.
Our experiments show that endogenous ERK5 translocates to nucleus in response to EGF stimulation by a mechanism driven by SUMO modification, since overexpression of ERK5 SUMO protease SENP2 effectively blocked this process (Figure 3). This is a classical experiment to demonstrate the role of a given SUMO protease (i.e. SENP2) in deSUMOylation of a given protein. In our case, our results clearly show SENP2 as the protease involved in switching off ERK5 SUMO modification. We could agree with the reviewer on that SENP2 overexpression could affect other proteins, but this also true for the suggested experiment using shRNAs. However, in our hands, overexpressed ERK5 forms recapitulate well what happens with the endogenous protein: activation (phosphorylation) in response to MEK5, and nuclear shuttling in response to MEK5 or Cdc37 overexpression (Figure 2, IF and subcellular fractionation experiements). Given the fact that the SUMO-deficient mutant does not enter the nucleus under any condition (even after inhibition of the nuclear export), we believe that our conclusions are well supported by the data presented in this work.
The same approach should be done in Fig. 5 experiments to show that SUMOylation is effectively required for ERK5 transcriptional activity and proliferation.
SUMO modification affects many proteins, including transcription factors that regulate cell proliferation (Rosonina, Transcription 8: 22031, 2017). Therefore, it has been reported that genetic manipulation of either SUMO proteins or PIAS ligases have a major impact in cancer cell proliferation Wang et al. Oncogene, 32: 2493-98, 2013; Han et al., Inter. J. Oncol. 52: 1081-109, 2018). Thus, silencing SUMO or SENP proteins do not allow us to delineate the exact role of ERK5 SUMOylation on cancer cell proliferation. For this precise reason, we performed proliferation experiments using in parallel the WT and the SUMO-deficient forms of ERK5. Our results show that overexpression of MEK5 and ERK5 induces PC-3 prostatic cancer cell proliferation (as shown by others, see McCracken et al., Oncogene 27: 2978-88, 2008), but no the SUMO deficient mutant (in spite of this mutant is phosphorylated and activated by MEK5).
Round 2
Reviewer 2 Report
I do not have additional comments to my previous remarks.